# Variable Selection for Multivariate Failure Time Data via Regularized Sparse-Input Neural Network

**DOI:** 10.3390/bioengineering12060596

**Published:** 2025-05-31

**Authors:** Bin Luo, Susan Halabi

**Affiliations:** 1School of Data Science and Analytics, Kennesaw State University, Kennesaw, GA 30144, USA; bluo@kennesaw.edu; 2Department of Biostatistics and Bioinformatics, Duke University, Durham, NC 27708, USA

**Keywords:** multivariate failure time, variable selection, high dimensionality, group LASSO, non-convex penalty

## Abstract

This study addresses the problem of simultaneous variable selection and model estimation in multivariate failure time data, a common challenge in clinical trials with multiple correlated time-to-event endpoints. We propose a unified framework that identifies predictors shared across outcomes, applicable to both low- and high-dimensional settings. For linear marginal hazard models, we develop a penalized pseudo-partial likelihood approach with a group LASSO-type penalty applied to the $\ell_{2}$ norms of coefficients corresponding to the same covariates across marginal hazard functions. To capture potential nonlinear effects, we further extend the approach to a sparse-input neural network model with structured group penalties on input-layer weights. Both methods are optimized using a composite gradient descent algorithm combining standard gradient steps with proximal updates. Simulation studies demonstrate that the proposed methods yield superior variable selection and predictive performance compared to traditional and outcome-specific approaches, while remaining robust to violations of the common predictor assumption. In an application to advanced prostate cancer data, the framework identifies both established clinical factors and potentially novel prognostic single-nucleotide polymorphisms for overall and progression-free survival. This work provides a flexible and robust tool for analyzing complex multivariate survival data, with potential utility in prognostic modeling and personalized medicine.

## 1. Introduction

Multivariate failure time data frequently arise in biomedical research, particularly when patients may experience multiple types of events or failures over the course of their diseases and treatments. For instance, in cancer studies, researchers may track various events such as time to recurrence, time to metastases, time to progression, and time to death for each patient. Identifying predictors associated with each failure time is crucial, especially in high-dimensional space when the number of potential covariates (*p*) far exceeds the sample size (*n*) in the presence of genetic and omics data [1,2,3,4]. Thus, effective variable selection methods play a pivotal role in distinguishing truly influential predictors from noisy variables and building parsimonious predictive models for multivariate failure times.

The predictor space in these biomedical studies is often heterogeneous, consisting of diverse data types. These include demographic characteristics (e.g., age, ethnicity), established clinical variables (e.g., disease stage, performance status, laboratory measures such as PSA or LDH), treatment-related variables, and increasingly, high-throughput omics data. The latter may comprise genetic variants such as single-nucleotide polymorphisms (SNPs), gene expression profiles, or proteomic markers, which can be continuous, binary, or categorical in nature. The complexity is further amplified when modeling multiple event types simultaneously, as associations between covariates and different failure outcomes may vary. In this setting, the extreme dimensionality of the data (p≫n) poses significant challenges for conventional modeling approaches, underscoring the need for robust and scalable variable selection techniques.

Our motivating example stems from the problem of identifying variables pertinent to two time-to-event endpoints, overall survival (OS) and progression-free survival (PFS), in men with advanced prostate cancer within a phase III clinical trial, where OS is typically defined as the time from randomization to death from any cause and PFS is defined as the time from randomization to disease progression or death from any cause, whichever occurs first. Alongside this trial, clinical factors and massive genetic data, including SNPs and plasma angiokines, were collected to gain a comprehensive understanding of the prognosis for both endpoints. In a metastatic disease state model where patients progress before death, the biological rationale suggests that factors predicting one outcome, such as PFS, are highly likely to predict the other outcome (OS). Given the strong correlation between OS and PFS, it naturally prompts the question: what are the most prognostic features associated with both endpoints? Instead of modeling each outcome individually, we seek to effectively determine the common predictors by integrating information from both endpoints. By identifying predictors for both outcomes, we enhance our understanding of prognoses, and these can be used in future trial design, thereby facilitating patient care and ultimately leading to improved clinical outcomes.

Various methodologies have been developed to identify relevant variables of outcomes, and these generally fall into three categories: filter [5,6,7], wrapper [8,9,10], and embedded methods [11,12,13,14]. Among them, penalized regression methods have gained a lot of traction since the introduction of the least absolute shrinkage and selection operator (LASSO) [11] in linear regression analysis. Examples of other commonly used penalized regression methods include the smoothly clipped absolute deviation (SCAD) [13], adaptive LASSO [12], and the minimax concave penalty (MCP) [14].

For survival data, adaptations and extensions of these penalized regression methods have been extensively explored for variable selection within the proportional hazards (PH) model [15,16,17,18]. For the analysis of multivariate failure data, researchers have proposed various methods including the penalized pseudo-partial likelihood method and the partially linear hazard regression model [19,20,21]. It is worth noting that these methods either assume a common parameter for each covariate across all types of outcomes or perform individual-level selection, thereby yielding varying sets of selected variables for each type of failure time. Recently, Cai et al. (2022) introduced a method grouping unknown parameters corresponding to the same predictor and conducting both group-level and individual-level selection [22]. However, their approach may still yield different selected variables for distinct types of outcomes. Furthermore, all the aforementioned penalized methods for multivariate failure time data impose a specific linear or partially linear form on the risk component within the hazard model. This entails a predefined structure for how covariates influence the hazard rate; however, the true relationship may be nonlinear and unknown in many real-world applications. For example, in biomedical studies, assuming a linear relationship might be overly restrictive, potentially failing to capture complex interactions between genetic markers and clinical variables, or non-monotonic effects common in biological systems, which could limit predictive accuracy.

To capture the complex relationship between predictors and time-to-event outcomes, Faraggi and Simon (1995) [23] proposed extending the PH model with a neural network. Katzman et al. (2018) [24] further investigated these models in the framework of deep learning, demonstrating that these novel networks could outperform the classical PH model in terms of the C-index. The application of advanced deep learning strategies continues to expand for tackling complex biomedical data challenges [25,26,27]. Recent advancements have introduced regularized neural networks for feature selection [28,29,30,31,32,33,34,35]. For instance, the group LASSO penalty has been applied to the weight vector associated with each node to induce sparsity among input nodes [29,30,31]. However, regularized neural networks incorporating LASSO tend to over-shrink the non-zero weight of relevant variables, leading to a high number of false positives in the selected model. To avoid the over-shrinkage of LASSO, Yamada et al. (2020) introduced stochastic gates to the input layer of neural networks and considered $l_{0}$-like regularization based on a continuous relaxation of the Bernoulli distribution [33]. More recently, Luo and Halabi (2023) [35] proposed a novel framework for sparse-input neural networks utilizing group concave regularization to address the limitations encountered with LASSO methods. However, these regularized neural net methods have not been developed yet for multivariate failure time data. Thus, there remains a critical need to develop methods that are capable of simultaneous variable selection and nonlinear function estimation for such data.

In this paper, we propose a novel framework designed for simultaneously selecting common variables and estimating models in the context of multivariate failure time data. Our approach starts by employing the penalized pseudo-partial likelihood method within the linear marginal hazard model. Specifically, we introduce a LASSO-type penalty to the $l_{2}$ norm of parameters associated with the same variables across all marginal hazard models. This group-level penalization facilitates the inclusion or exclusion of the entire group of parameters, enabling the identification of common variables relevant to multivariate failure time data. To overcome the potential limitations of linearity, we then significantly extend this framework to handle potentially complex, nonlinear relationships using sparse-input feed-forward neural networks, where each marginal hazard function is approximated by such a network with an identical input layer. Within the framework of the penalized pseudo-partial likelihood, we treat all outgoing connections from a single input neuron across feed-forward neural networks as a collective group and apply a proper penalty to the $l_{2}$ norm of weights within each group. By shrinking the weights of specific groups to exact zeros, our method yields a collection of neural networks utilizing only a concise subset of common variables. We validate our proposed approach through simulation studies and real data examples, demonstrating its satisfactory finite sample performance for bivariate failure time data. Furthermore, we show that the proposed framework effectively handles scenarios where the influential predictor sets only partially overlap between outcomes, enhancing its practical applicability.

The subsequent sections of this article are organized as follows: In Section 2, we define the problem of feature selection within the framework of the marginal proportional hazards model and introduce our innovative approach for both linear and nonlinear hazards models. In Section 3, we provide the details of implementation, including the optimization algorithm and the procedure for selecting tuning parameters. In Section 4, we present extensive simulation studies to demonstrate the performance of the proposed methodology. We apply the proposed method to a real-world dataset in Section 5. In Section 6, we discuss the results and their implications. Finally, Section 7 concludes the paper by summarizing our main findings and their broader significance.

## 2. Methodology

Suppose that there are *J* types of failure modes for each of *n* independent individuals. Denote $T_{ij}$ and $C_{ij}$ as the survival and censoring time for the *j*th type of failure of the *i*th individual, respectively, for i=1,…,n and j=1,…,J. Define $Y_{ij}$ as the observed time $=\min(T_{ij},C_{ij})$, and $\delta_{ij}$ as the censoring indicator $=I(T_{ij}\leq C_{ij})$. Let $X_{i}={\left(x_{i1},\ldots ,x_{ip}\right)}^{T}$ denote a *p*-dimensional covariate vector for the *i*th individual. For the *j*th type of failure of the *i*th individual, we assume the marginal hazard function of $T_{ij}$ to take the following form of the proportional hazards model:(1)$$h_{j}\left(t|X_{i}\right)=h_{0j}\left(t\right)\exp\left\{f_{j}\left(X_{i};w_{j}\right)\right\},$$
where $f_{j}$ denotes the type-specific risk function, $w_{j}$ is a vector of unknown parameters, and $h_{0j}$ is an unspecified type-specific baseline hazard function. In addition, we incorporate the sparsity assumption for the marginal hazard model, meaning that the risk function $f_{j}$ depends on a common subset of covariates S⊆{1,…,p} for all *J* types of time-to-event outcomes. In our specific scenario with bivariate failure times with J=2, for instance, this implies the presence of a common set of prognostic factors that are predictive for both OS and PFS.

### 2.1. Marginal Linear Hazards Model

When the risk function $f_{j}$ (within the exponent) takes a linear form, represented as $f_{j}\left(X_{i};w_{j}\right)=w_{j}^{T}X_{i}$ with $w_{j}={\left(w_{j1},\ldots ,w_{jp}\right)}^{T}$, the model described in Equation (Equation 1) corresponds to the standard marginal hazard model, as previously investigated in studies [36,37,38]. These studies estimated the unknown vector of parameters $w_{j}$ by minimizing the negative pseudo-partial likelihood function:$${\tilde{L}}_{n}\left(w\right)=-\sum_{i=1}^{n}\sum_{j=1}^{J}\delta_{ij}\left\{w_{j}^{T}X_{i}-\log\left(\sum_{k\in R_{ij}}\exp\left(w_{j}^{T}X_{k}\right)\right)\right\},$$
where $w\in R^{pJ}$ is the column-vector concatenation of all parameters in $\left\{w_{j}:j=1,\ldots ,J\right\}$, and $R_{ij}=\left\{k:Y_{kj}\geq Y_{ij}\right\}$ represents the set of individuals at risk just before the observed time $Y_{ij}$ for the *j*th failure of the *i*th observation. To simultaneously select the important features for all *J* types of failure and estimate parameters, we introduce the following group-regularized negative pseudo-partial likelihood function:(2)$$\tilde{w}=\underset{w\in R^{pJ}}{\mathrm{argmin}}\left\{{\tilde{L}}_{n}\left(w\right)+\sum_{k=1}^{p}\rho_{\lambda }(∥W_{k}{∥}_{2})\right\},$$
where $W_{k}={\left(w_{1k},\ldots ,w_{Jk}\right)}^{T}$ denotes the parameter vector of the *k*th covariate across *J* types of outcomes. The penalty function $\rho_{\lambda }$, parameterized by λ≥0, plays a crucial role. By grouping regression parameters associated with the same covariate, $\rho_{\lambda }$ effectively shrinks specific groups of parameter vectors, $W_{k}$, to precisely zero, resulting in models that rely on a consistent, smaller set of variables. Figure 1a shows the network architecture for the marginal linear hazards model. Several penalty functions have been commonly employed to promote sparsity in the solution, including LASSO [11], SCAD [13], and MCP [14]. When applied to the $l_{2}$-norm of the parameter vector for certain groups, these penalty functions result in group regularization techniques such as group LASSO (GLASSO) ([39]), group SCAD (GSCAD) ([40]), and group MCP (GMCP) [41]. Below are the expressions of LASSO and MCP, which will be considered in our numerical analyses.
**LASSO**$$\rho_{\lambda }\left(t\right)=\lambda \left|t\right|.$$**MCP**$$\rho_{\lambda }\left(t\right)=\mathrm{sign}\left(t\right)\lambda \int_{0}^{\left|t\right|}{\left(1-\frac{z}{\lambda a}\right)}_{+}dz,$$
where a>0 is fixed.
A folded concave penalty, such as the SCAD or MCP, has been shown to have better variable selection properties than a convex penalty, including the LASSO [13,14].

### 2.2. Marginal Nonlinear Hazards Model

While assuming a linear relationship between the covariates and the outcomes may offer convenience, in real applications, the actual relationship can be complex and the form of the risk function $f_{j}$ may not be readily available. In our second approach, we address this complexity by approximating $f_{j}$ using neural networks. Specifically, for the *j*th type of failure, we consider the feed-forward neural network, denoted as $f_{w_{j}}:R^{p}\mapsto R$ parameterized by $w_{j}$, to approximate $f_{j}$. Consequently, the marginal hazard model takes the form of $h_{j}\left(t|X_{i}\right)=h_{0j}\left(t\right)\exp\left\{f_{w_{j}}\left(X_{i}\right)\right\}$. The negative pseudo-partial likelihood function becomes$$L_{n}\left(w\right)=-\sum_{i=1}^{n}\sum_{j=1}^{J}\delta_{ij}\left\{f_{w_{j}}\left(X_{i}\right)-\log\left(\sum_{k\in R_{ij}}\exp\left(f_{w_{j}}\left(X_{k}\right)\right)\right)\right\}.$$We propose to simultaneously select common variables for *J* types of failures and train the neural networks by minimizing the following group-regularized negative pseudo-partial likelihood function:(3)$$\hat{w}=\underset{w\in R^{d}}{\mathrm{argmin}}\left\{L_{n}\left(w\right)+\sum_{k=1}^{p}\rho_{\lambda }(∥W_{0,k}{∥}_{2}{)+\alpha ∥w∥}_{2}^{2}\right\},$$
where $w\in R^{d}$ denotes the column vector formed by concatenating all parameters from $\left\{w_{j}:j=1,\ldots ,J\right\}$, and *d* is the total number of elements in w, including all weights and biases from the *J* neural networks. Here, $W_{0,k}$ represents the subvector of parameters w and denotes the weight vector associated with the *k*th covariates in the input layers of the neural networks $f_{w_{j}}$ for *J* types of failures. The main concept involves applying an appropriate group penalty to the $l_{2}$ norm of weights from all outgoing connections of each input node, thereby producing a set of sparse-input neural nets that utilize only a limited subset of the original variables. Figure 1b shows the network architecture of the marginal nonlinear hazards model. It is important to highlight that the ridge regularization term with α>0 is also included in Equation (Equation 3) to prevent over-fitting in the neural network. Let $\hat{S}=\{k:∥{\hat{W}}_{0,k}{∥}_{2}\neq 0\}$ be the index set of selected variables. Consequently, we can obtain the model fit for the *j*th failure time, represented by the estimated risk function $f_{{\hat{w}}_{j}}$, which depends on the same subset of variables $\hat{S}$ across *J* types of failures.

It is noteworthy that by removing the hidden layers of the neural network $f_{w_{j}}$, i.e., the blue boxes in Figure 1b, $f_{w_{j}}$ becomes a linear function of $X_{i}$. In this case, the proposed framework in Equation (Equation 3) with α=0 simplifies to the estimator in Equation (Equation 2). Additionally, when J=1, our approach is equivalent to the sparse-input neural network framework proposed in Luo and Halabi (2023) [35] for survival analysis with a single outcome.

## 3. Implementation

In this section, we detail the implementation of the proposed method. As previously described, the estimator in Equation (Equation 2) for the linear hazards model is a special case of Equation (Equation 3). Therefore, our attention is drawn to the optimization problem in Equation (Equation 3). We use the composite gradient descent algorithm to solve for critical points for the objective function in Equation (Equation 3)  [42]. Although the objective function is non-convex, the approach described by Gong (2013) [43] enables us to apply composite gradient descent to non-convex optimization. The core algorithm aligns with the computational framework established in Luo and Halabi (2023) [35], with the adaptation extending the loss function $L_{n}$ to accommodate the multivariate failure time data.

We follow a two-step process in training the sparse-input neural networks in Equation (Equation 3). Initially, a standard gradient descent step with learning rate γ is applied to the smooth component of the objective function ${\bar{L}}_{n,\alpha }\left(w\right)=L_{n}\left(w\right)+{\alpha ||w||}_{2}^{2}$, considering all weight parameters. The learning rate γ>0 can be set as a fixed value or determined by employing the backtracking line search method, as described in Nesterov (2013) [42]. Subsequently, a proximal operator is employed on the weight parameters of the input layer. This sequential approach makes the procedure exceptionally straightforward to implement within prevalent machine learning frameworks. The outlined procedure is summarized in Algorithm 1. Here $A_{j}$ represents the index set of $W_{0,j}$ within w. Notably, the symbol PROX denotes the proximal operator determined by the form of the penalty function $\rho_{\lambda }$. Choosing $\rho_{\lambda }$ as the LASSO and MCP penalty, it can be verified that the proximal operator takes the following form:LASSO$$PROX_{\mathrm{LASSO}}\left(z;\gamma ,\lambda \right)=S_{g}\left(z,\gamma \lambda \right).$$MCP$$PROX_{\mathrm{MCP}}\left(z;\gamma ,\lambda \right)=\left\{\begin{array}{cc} \frac{a}{a-\gamma }S_{g}\left(z,\gamma \lambda \right), & {\mathrm{if}\;∥z∥}_{2}\leq a\lambda ,\\ z, & {\mathrm{if}\;∥z∥}_{2}>a\lambda , \end{array}\right.$$where $S_{g}\left(z;\lambda \right)$ is the group soft-thresholding operator, defined as$$S_{g}\left(z;\lambda \right)={\left(1-\frac{\lambda }{{∥z∥}_{2}}\right)}_{+}z.$$
**Algorithm 1:** Training the multivariate regularized sparse-input neural network**Input:** Training dataset ${\left\{X_{i},Y_{ij},\delta_{ij}\right\}}_{i=1}^{n}$, feed-forward neural network $f_{w_{j}}\left(\cdot \right)$ for j=1,…,J, increasing sequence of tuning parameters ${\left\{\lambda_{k}\right\}}_{k=1}^{m}$, regularization parameter α, learning rate γ, number of epochs *B***Output:** Solution path of weight parameters ${\left\{w^{k}\right\}}_{k=1}^{m}$ in trained neural networksInitialize model parameter w, and set k=1**while** 
k≤m 
**do**    **for** b=1 **to** *B* **do**        Compute gradient of the smooth loss ${\bar{L}}_{n,\alpha }\left(w\right)$ using back-propagation        Update $w\leftarrow w-\gamma \nabla {\bar{L}}_{n,\alpha }\left(w\right)$        **for** j=1**to***p* **do**           Update $w_{A_{j}}\leftarrow \mathrm{Prox}\left(w_{A_{j}};\gamma ,\lambda_{k}\right)$        **end for**    **end for**    Update $w^{k}\leftarrow w$    Increment *k***end while**

Algorithm 1 employs a backward path-wise optimization strategy, starting with a dense model at the minimum value of $\lambda_{1}$ ($\lambda_{\min}$) and progressing towards sparse models up to $\lambda_{m}$ ($\lambda_{\max}$), excluding all variables from the network. This warm start strategy, transitioning from dense to sparse, is also implemented in Luo and Halabi (2023) utilizing group concave regularization and has demonstrated the capability to yield a seamless transition from dense to sparse models [35].

In addition to ensuring stable and smooth solution paths, the backward path-wise optimization approach offers computational advantages. The successive weight estimates along the path tend to exhibit similarity, thereby reducing the number of gradient descent iterations required per step. This concentration of computations primarily at $\lambda_{\min}$ enhances computational efficiency. Moreover, variables excluded from previous solutions are rarely reintroduced in subsequent solutions, leading to pruning along the path and reducing complexity as the model becomes sparse. This results in a substantial acceleration of computation, particularly beneficial for high-dimensional data.

Regarding implementation details, for the nonlinear marginal hazard models, where each risk function $f_{w_{j}}$ is approximated by a neural network, we adopted a standardized architecture across all numerical studies to ensure fair comparisons. Specifically, we employed a multi-layer perceptron (MLP) with two hidden layers, each comprising 10 units and using rectified linear unit (ReLU) activation functions. Network weights were initialized from a Gaussian distribution with a mean of zero and standard deviation of 0.1, while bias terms were set to zero, to ensure a consistent starting point for optimization and to help break symmetry in the network. Additional details regarding scenario-specific settings, such as the number of training epochs and the ranges of hyperparameters considered, are provided in the Appendix A, Section SA.

The proposed framework requires the tuning of two key hyperparameters: the group penalty coefficient λ, which governs model sparsity, and the ridge penalty coefficient α, which mitigates the risk of overfitting. In our numerical study, we reserved a 20% validation set from the training data. The model was then trained on the remaining 80%, exploring a fine grid of λ and α values. The optimal combination was selected as the one minimizing the negative log pseudo-partial likelihood evaluated on the validation set. This hyperparameter tuning procedure was applied independently for each training dataset in both the simulation replications and the real-data analysis. The final model performance was then assessed on a separate, unseen test set. Full details of this procedure are provided in Section 4 and Section 5.

The computational complexity of our proposed method is primarily determined by the number of samples (*n*), the total number of initial predictors (*p*), the number of failure types (*J*), the fixed neural network architecture specified for each outcome branch, and the parameters associated with path-wise optimization—namely, the number of regularization parameters (*m*) and the number of training epochs per λ value (*T*). The approximate overall complexity of computing the full solution path across all *J* outcomes can be characterized as $O(Jn\bar{d}Tm)$, where $\bar{d}$ denotes the average number of active (i.e., nonzero) input features along the solution path. Note that the use of warm starts and an aggressive feature pruning strategy—often resulting in $\bar{d}\ll p$ under high-dimensional sparsity—yields substantial computational savings in practice.

Python code and examples for the proposed multivariate regularized neural networks are available at https://github.com/r08in/GCRNN, accessed on 1 May 2025.

## 4. Simulation Studies

We evaluate the finite sample performance of the proposed regularized neural networks for feature selection and prediction through simulation studies. We simulate J=2 failure types with bivariate survival time $\left(T_{1},T_{2}\right)$ generated from the multivariate Clayton and Cuzick distribution [44], where the joint survival function is$$S\left(t_{1},t_{2};X\right)={\left\{\sum_{j=1}^{2}S_{j}{\left(t_{j};X\right)}^{-\theta }-1\right\}}^{-\frac{1}{\theta }},$$
where $S_{j}$ is the marginal survival function for the *j*th type of failure, which can be obtained from the marginal proportional hazards model in Equation (Equation 1). The parameter θ>0 controls the dependence among survival time $\left(T_{1},T_{2}\right)$ and is related to Kendall’s τ via τ=θ/(θ+2). Thus, θ→∞ yields an increasing positive correlation, and θ=0 corresponds to independence. We set θ to 0.01, 2, and 8 and the corresponding Kentall’s τ=0.005,0.5, and 0.8, respectively.

Specifically, the covariate vectors $X={\left(X_{1},\ldots ,X_{p}\right)}^{T}\in R^{p}$ are generated from the independent standard normal distribution. Thus, $T_{j}=H_{0j}^{-1}\left(-\log\left(U_{j}\right)\exp(f_{j}\left(X_{i}\right)\right)$ for j=1,2, where $H_{0j}$ is the baseline cumulative hazard function for the *j*th failure type, which is defined as $H_{0j}\left(t\right)=\int_{0}^{t}h_{0j}\left(u\right)du$, and $\left(U_{1},U_{2}\right)$ are bivariate random variables with marginal uniform distribution in [0,1] and joint distribution function given by a Clayton copula $C\left(u_{1},u_{2}\right)={\left(u_{1}^{-\theta }+u_{2}^{-\theta }-1\right)}^{-\frac{1}{\theta }}$. In our simulation, we considered a Weibull hazard function for $H_{0j}$, with the scale parameter $s_{j}$ ($s_{1}=s_{2}=2$) and the shape parameter $k_{j}$ ($k_{1}=1$ and $k_{2}=2$). For the *j*th failure type, we simulate data for *n* individuals. A proportion C of the individuals are randomly selected to be censored, with censoring indicators defined as $\delta_{j}=1$ for event occurrences and $\delta_{j}=0$ for censored observations. The observed time is given by$$Y_{j}=T_{j}I\left(\delta_{j}=1\right)+C_{j}I\left(\delta_{j}=0\right),$$
where $T_{j}$ denotes the event time and $C_{j}$ is the censoring time. For censored individuals, $C_{j}$ is independently drawn from a uniform distribution on $\left(0,T_{j}\right)$, ensuring that the censoring occurs prior to the event. We generate *n* i.i.d. random samples with various forms of the risk function $f_{j}$ in the following examples:

**Example 1.** **(Linear Hazards Model).** 
*We consider a linear form of the risk function as follows:*

$$f_{j}\left(X_{i}\right)=X_{i}^{T}\beta_{j},$$



*where $\beta_{1}={\left(0.729,0.493,0.200,-1.625,-1.081,\underset{p-5}{\underset{︸}{0,\ldots ,0}}\right)}^{T}$,*

*and $\beta_{2}={\left(0.701,-1.095,0.435,0.080,0.053,\underset{p-5}{\underset{︸}{0,\ldots ,0}}\right)}^{T}$. Here, the non-zero components of $\beta_{j}$ are independently generated from a standard Gaussian distribution. We consider a low-dimensional setting in this example with p=20, n=300,500, and the censoring rate C=0.1 and 0.3.*



**Example 2.** **(Nonlinear Hazards Model).**
*We consider the nonlinear form of the risk function in the following two models:*


*Nonlinear Model 1 (NLM1)*

$$\begin{array}{cc} f_{1}\left(X_{i}\right) & =3X_{i1}+4X_{i1}X_{i2}+X_{i1}^{2}+5X_{i2}^{2}+X_{i3}^{2}+3X_{i4}+5X_{i4}X_{i5}\\ f_{2}\left(X_{i}\right) & =X_{i1}^{2}+2X_{i2}+3X_{i3}+4X_{i3}X_{i2}+5X_{i5}+3X_{i5}X_{i4} \end{array}$$


*Nonlinear Model 2 (NLM2)*

$$\begin{array}{cc} f_{1}\left(X_{i}\right) & =X_{i1}+X_{i1}X_{i2}+\log(|X_{i2}\left|+0.1\right)+\exp\left(X_{i3}+X_{i4}\right)+X_{i5}^{2}\\ f_{2}\left(X_{i}\right) & =X_{i1}^{2}+4X_{i2}+X_{i2}X_{i3}+3\log(|X_{i3}\left|+0.5\right)+2\exp\left(X_{i4}+X_{i5}\right) \end{array}$$



*We consider low-dimensional (p=20) and high-dimensional (p=1000) scenarios in this example with n=500 and C=0.2.*


We generated 200 replicates for each simulation configuration. For each replicate, the proposed models were first trained on a generated dataset of *n* individuals. During this training process, optimal hyperparameters (such as λ and α) were selected using the validation set methodology detailed in Section 3, where 20% of the generated training data were reserved for this tuning purpose. The final tuned model’s performance in prediction and feature selection was then rigorously evaluated on a separate, independently generated test set, also comprising *n* new random samples. This test set was drawn from the same underlying data-generating distribution as the training data but was not used during any phase of model training or hyperparameter selection. The following measures, calculated on this independent test set, were used to assess performance:(1)The concordance index (C-index) is a standard metric that evaluates the predictive ability of the learned model in survival analysis by measuring the agreement between the ranking of the predicted and observed survival times. The C-index is reported for each type of failure.(2)False positive rate (FPR) is the percentage of selected but unimportant covariates:$$FPR=\frac{|\hat{S}⋂S^{c}|}{|S^{c}|}\times 100\%.$$(3)False negative rate (FNR) is the percentage of non-selected but important covariates:$$FNR=\frac{|{\hat{S}}^{c}⋂S|}{\left|S\right|}\times 100\%.$$(4)Identical rate (IR) is the percentage of replicates where the exact same set of indices is selected across *J* failure types. For methods from the proposed frameworks, IR =100%.(5)Model size (MS) is the average number of selected covariates.

For this simulation, the true set of indices corresponding to important variables is S={1,2,3,4,5}. The set of indices for variables selected by the model is denoted by $\hat{S}=\{k:∥{\hat{W}}_{0,k}{∥}_{2}\neq 0\}$, as previously defined.

In our numerical studies, we utilize the MCP penalty in our proposed framework, as both the SCAD penalty and the MCP penalty perform similarly in general. For the analysis of bivariate failure time data, we refer to the method employing the group LASSO and group MCP for the linear marginal hazards model as Bi-LASSO and Bi-MCP, respectively. Additionally, we term the methods based on neural networks as Bi-GLASSONet and Bi-GMCPNet, respectively.

In Example 1 of linear hazards models, we compare the proposed estimator Bi-MCP with Bi-LASSO due to the lack of existing methods for simultaneous parameter estimation and selection of the same set of variables across different types of failures. To illustrate the advantages of using multivariate failure time data, we also include the standard MCP and LASSO methods applied to each type of failure for comparison. Moreover, we consider the oracle Cox’s estimator as a benchmark, where the true relevant variables are known in advance. For Example 2, which involves the nonlinear hazards models, we compared the proposed method Bi-GMCPNet with Bi-GLASSONet. In addition, we compare GMCPNet [35] for single failure time data and the oracle neural networks (Oracle-NN) method using only the relevant variables as the benchmark. We also include random survival forests (RSF) for modeling each outcome separately. See the Appendix A for the implementation details.

### 4.1. Results

Table 1 provides a comprehensive overview of the feature selection performance in Example 1. Across all scenarios for the linear hazards model, Bi-MCP consistently outperforms other methods in terms of feature selection, demonstrating lower FPR and FNR. It is important to highlight that Bi-LASSO tends to select more variables, resulting in larger model sizes across all settings. Additionally, both the standard MCP and LASSO exhibit relatively higher FPR and FNR for either one or both of the outcomes. In contrast, Bi-MCP enhances selection performance by incorporating information from bivariate failure times. Notably, the IR values for MCP and LASSO are nearly zero across all settings, indicating that the separately fitted models rarely select identical sets of variables. While they may capture overlapping predictors—including the true shared ones—their selections diverge considerably, likely due to stochastic variation in the inclusion of noise variables.

Figure 2 presents the predictive performance of each method under the linear hazards model (Example 1), based on 200 simulation replicates with a censoring rate of 30% and a sample size of n=300. Across varying levels of dependence between event times (Kendall’s τ), the figure illustrates how well each approach distinguishes event orderings, as measured by the C-index. Similar results for other settings are observed as indicated in the Appendix A. Bi-LASSO shows the lowest C-index for both survival outcomes, primarily due to issues related to over-selection. Conversely, Bi-MCP slightly outperforms MCP and LASSO due to more accurate variable selection and proves comparable to oracle Cox’s estimator in most scenarios. These results remain consistent across different censoring proportions and Kendall’s τ for the strength of correlation of bivariate failure times.

Focusing on the nonlinear hazards models in Example 2, Table 2 provides insights into the feature selection results. In low-dimensional settings (p=20), both Bi-GMCPNet and GMCPNet yield model sizes closely aligned with the true model, displaying low FPR and FNR across most scenarios. Bi-GMCPNet slightly outperforms GMCPNet by simultaneously analyzing bivariate failure times. Conversely, Bi-GLASSONet encounters challenges with over-selection, mirroring the results in Example 1. In high-dimensional settings (p=1000), while GMCPNet tends to miss some important variables, resulting in higher FNR for both outcomes, our proposed Bi-GMCPNet maintains high accuracy in variable selection.

Figure 3 summarizes the performance of each method under nonlinear bivariate failure time models (Example 2) with Kendall’s τ=0.5, across both low- and high-dimensional settings. In low-dimensional settings (p=20), Bi-GMCPNet exhibits performance similar to that of GMCPNet, and both are comparable to Oracle-NN. However, in high-dimensional settings with p=1000, GMCPNet experiences a decline in prediction accuracy, yielding lower C-index values overall. In contrast, Bi-GMCPNet maintains performance comparable to oracle-NN, aligning with their variable selection capabilities. Notably, the issue of overselection in Bi-LASSO results in the poorest predictive performance. In addition, RSF consistently performs poorly compared to most other methods due to its inability to handle multivariate data and variable selection.

In summary, the results of our simulation underscore the superior performance of the proposed framework for multivariate failure time data in feature selection and prediction. Bi-MCP and Bi-GMCPNet enhance variable selection by incorporating multiple types of failure and provide consistent variable selection across different outcomes. The proposed framework shows promise in addressing the challenges of feature selection and prediction in high-dimensional data. Notably, as the tuning parameters are selected based on predictive performance on a validation set, the high FPR observed for Bi-LASSO and Bi-GLASSONet suggests that the LASSO penalty tends to include many false positives to offset its inherent bias, thereby enhancing predictive accuracy at the expense of model sparsity.

### 4.2. Relaxing the Common Predictor Assumption

To assess the robustness of the proposed methods when the assumption of a shared set of important predictors across all failure types is violated, we consider Example 3, a variant of the nonlinear model 2 (NLM2) used in Example 2.

**Example 3.** **(Nonlinear Hazards Model with Different Predictor Sets).** 
*We retain the functional form of $f_{1}$ but modify $f_{2}$ to depend on a different subset of covariates. Specifically, we replace $X_{4}$ and $X_{5}$ in $f_{2}$ with $X_{6}$ and $X_{7}$, yielding the following risk functions:*

$$\begin{array}{cc} f_{1}\left(X_{i}\right) & =X_{i1}+X_{i1}X_{i2}+\log(|X_{i2}\left|+0.1\right)+\exp\left(X_{i3}+X_{i4}\right)+X_{i5}^{2}\\ f_{2}\left(X_{i}\right) & =X_{i1}^{2}+4X_{i2}+X_{i2}X_{i3}+3\log(|X_{i3}\left|+0.5\right)+2\exp\left(X_{i6}+X_{i7}\right) \end{array}$$

*In this setup, the true predictor set for the first failure type is $S_{1}=\left\{1,2,3,4,5\right\}$, and for the second failure type, $S_{2}=\left\{1,2,3,6,7\right\}$. The set of predictors common to both outcomes is $S_{\mathrm{common}}=S_{1}\cap S_{2}=\left\{1,2,3\right\}$, while the union of all relevant predictors is $S_{\mathrm{union}}=S_{1}\cup S_{2}=\left\{1,2,3,4,5,6,7\right\}$. We fix Kendall’s τ=0.5 to reflect moderate dependence between failure times and examine both low-dimensional (p=20) and high-dimensional (p=1000) settings with sample size n=500 and censoring rate C=0.2.*


We compare our proposed Bi-GMCPNet with Bi-GLASSONet and the single-outcome GMCPNet applied separately to each failure type.

Table 3 presents the results. In the low-dimensional case (p=20), Bi-GMCPNet identifies a model with average size 7.1, aligning closely with $|S_{\mathrm{union}}|=7$. This indicates that the method successfully captures all predictors relevant to at least one failure type. From the union perspective, this yields zero false negatives (FNR = 0%), although the inclusion of outcome-specific predictors leads to a slightly elevated false positive rate (FPR = 14%) relative to the minimal set for a single outcome (size 5). Importantly, the method maintains strong predictive performance, as reflected in high C-index values. By contrast, Bi-GLASSONet selects all variables indiscriminately (FPR = 100%), demonstrating poor variable selection—consistent with its behavior observed in Example 2. Meanwhile, GMCPNet applied individually selects models closer to the size of the separate true sets ($S_{1}$ and $S_{2}$).

In the high-dimensional setting (p=1000), the advantages of Bi-GMCPNet are even more pronounced. It selects a parsimonious model (average size 6.5), again approximating the true union set, with minimal FPR (0.1–0.2%). It successfully recovers the shared predictors and most of the outcome-specific ones, yielding false negative rates (FNR) of 11.2% for outcome 1 and, notably, 0% for outcome 2. In contrast, Bi-GLASSONet performs poorly, selecting an excessive number of variables (average model size = 265.4), with high FNRs and low predictive accuracy (as measured by the C-index). Compared to running GMCPNet separately for each outcome, Bi-GMCPNet achieves comparable or slightly better predictive performance (C-index: 0.820 and 0.923 vs. 0.820 and 0.910), while significantly improving variable selection. The separate applications of GMCPNet result in higher FNRs (13.5% and 14.7%), whereas Bi-GMCPNet, by leveraging its joint modeling framework, effectively borrows strength across outcomes—substantially reducing the FNR for outcome 2 and modestly improving it for outcome 1. This demonstrates its superior ability to recover relevant predictors, even when the sets are not completely shared.

These results illustrate the robustness of Bi-GMCPNet to violations of the common predictor assumption. Its group-penalization strategy implicitly encourages selection of the union of important predictors ($S_{1}\cup S_{2}$), rather than restricting to the intersection ($S_{1}\cap S_{2}$). This behavior is practically beneficial, as it yields a unified model that retains all relevant prognostic variables across failure types while remaining sparse and interpretable. In contrast to Bi-GLASSONet and the outcome-specific GMCPNet, Bi-GMCPNet offers superior selection and prediction performance, even in challenging high-dimensional settings with only partially overlapping predictor sets. These findings highlight its utility as a robust and unified approach for analyzing multivariate failure time data.

## 5. Real Data Example

We leverage data from the CALGB-90401 study [45], a double-blinded phase III clinical trial comparing docetaxel and prednisone with or without bevacizumab in men with metastatic castration-resistant prostate cancer (mCRPC), to demonstrate the efficacy of our proposed method. The CALGB-90401 dataset comprises 498,801 single-nucleotide polymorphisms (SNPs) derived from blood samples of patients. Our primary aim is to develop a single model for both OS and PFS utilizing the SNPs and clinical data, with a focus on 625 DNA damage repair genes [46,47,48,49,50,51,52]. Adopting a dominant model, each SNP is treated as a binary variable. Additionally, we incorporate eight clinical variables recognized as established prognostic markers for overall survival in mCRPC patients [53]: opioid analgesic use (PAIN), ECOG performance status, albumin (ALB), disease site (classified as lymph node only, bone metastases with no visceral involvement, or any visceral metastases), LDH greater than the upper limit of normal (LDH.High), hemoglobin (HGB), PSA, and alkaline phosphatase (ALKPHOS). The final dataset consists of p=635 variables, with n=631 patients and censoring rates of 6.8% for OS and 6.3% for PFS.

We adopt the marginal proportional hazards model represented by Equation (Equation 1) for our proposed methods, aiming to identify clinical variables or SNPs predictive of both OS and PFS in patients with mCRPC. To assess the performance and stability of our proposed methods on real-world data, we employed a rigorous internal validation strategy based on repeated random subsampling (i.e., Monte Carlo cross-validation). Specifically, the complete CALGB-90401 dataset (n=631) was randomly partitioned 100 times into training (83%, n≈526) and testing (17%, n≈105) subsets. For each of the 100 iterations, the following procedure was applied:Model Training and Hyperparameter Tuning: Models were trained exclusively on the training data. Within each training set, hyperparameters (λ and α) were selected using the validation approach described in Section 3. This involved further splitting the training data into an 80% fitting set and a 20% internal validation set, with the optimal parameters chosen based on the validation-set performance.Performance Evaluation: The predictive performance of the tuned model was then evaluated on the held-out testing set using the time-dependent area under the receiver operating characteristic curve (tAUC), calculated via Uno’s estimator [54]. The tAUC is a summary measure that captures how well each model discriminates between patients who experience an event versus those who do not, across all time points.

Figure 4 presents the evaluation of model performance on the CALGB-90401 advanced prostate cancer dataset using 100 random train/test splits. The left and middle panels summarize predictive accuracy for overall survival (OS) and progression-free survival (PFS), respectively, based on the tAUC. In predicting OS (left panel), our proposed bivariate methodology, Bi-MCP (linear model) and Bi-GMCPNet (nonlinear model) outperform their respective single-variate counterparts, MCP and GMCPNet. Notably, Bi-GMCPNet demonstrates superior performance across all methods, as indicated by higher tAUC in overall survival prediction. The efficacy of Bi-GMCPNet is particularly pronounced for PFS (middle panel), indicating the importance of integrating information from multivariate outcomes in the processes of both feature selection and modeling. It is worth noting that the Bi-GLASSONet method exhibits the lowest predictive performance, displaying a tendency to over-select variables, which is consistent with our simulation results.

The right panel of Figure 4 displays the consistency of variable selection by Bi-GMCPNet, showing clinical features and SNPs selected in at least 10% of the data splits. These results highlight the model’s ability to identify most of the clinically significant variables and select crucial SNPs in the prediction of both OS and PFS. Notably, seven out of eight clinical variables previously established as prognostic markers for overall survival in mCRPC were selected with high proportions by our model [53]. This alignment highlights the model’s capability to robustly identify clinically meaningful predictors. Among the selected SNPs, several have been linked to prostate cancer progression and mortality, particularly those involved in DNA repair and immune regulation. For instance, rs12757998 in RNASEL, a gene critical for antiviral defense and apoptosis, has been associated with high-grade prostate cancer and may influence response to radiation therapy (Meyer et al., 2010 [55]). SNPs in DNA repair genes—such as rs8058895 in FANCA, rs623860 in ATM, and rs1981929 in MSH2—have been implicated in increased risk of aggressive prostate cancer and worse clinical outcomes, likely due to impaired DNA damage response (Pritchard et al., 2016 [56]; Na et al., 2017 [57]; Mateo et al., 2020 [58]). Similarly, rs12112229 in PMS2, another mismatch repair gene, has been linked to prostate cancer recurrence and progression (Norris et al., 2009 [59]; Fukuhara et al., 2015 [60]). rs8075760 in RAD51D, which plays a role in homologous recombination, has also been associated with elevated cancer risk, including ovarian cancer (Loveday et al., 2011 [61]). While these variants show compelling associations with disease severity and mortality, other SNPs—such as rs1011019 (ERCC3), rs13149290 (ZNF827), rs6470494 (PCAT2), rs368248 (SLC8A1), and rs12155172 (LINC01162)—have limited or emerging evidence regarding their role in prostate cancer death and require further investigation. These genetic markers offer valuable insight into the biological underpinnings of prostate cancer and may serve as potential prognostic tools for identifying patients at higher risk of lethal disease.

## 6. Discussion

This paper has introduced a novel regularized framework designed for simultaneous variable selection and flexible model estimation in the complex setting of multivariate failure time data. We have presented approaches for both linear and nonlinear marginal hazard models, leveraging group-level penalization to identify common predictors. The preceding sections detailed the methodology and demonstrated its performance through extensive simulations and a real-world application. In this section, we will further elaborate on the significance and interpretation of these findings, discuss the advantages and robustness of our proposed methods in comparison to alternatives, address their potential limitations, and outline promising avenues for future research.

The results of our simulation studies clearly demonstrate several key advantages of our proposed framework, particularly when compared against common alternative approaches or methods that do not jointly model multiple outcomes. For linear hazard models, the proposed Bi-MCP outperformed methods applied to individual outcomes by showing lower FPR and FNR. This approach successfully identified common prognostic factors for both time-to-event outcomes, demonstrating its robustness. Furthermore, for nonlinear hazard models, Bi-GMCPNet exhibited superior performance in capturing complex relationships between covariates and outcomes. It outperformed classical linear models, which are inherently unable to capture such nonlinearities, as well as standard neural network approaches when applied to individual outcomes without joint modeling. This capability of Bi-GMCPNet to utilize neural networks for nonlinear function approximation, while simultaneously enforcing sparsity among input features through group-level penalties, proved highly effective and addresses a notable gap where many existing penalized methods for multivariate failure time data are restricted to linear assumptions. Such group-level penalties not only simplified the models but also ensured that the selected variables were consistently relevant across multiple outcomes.

Regarding the performance of Bi-GLASSONet in high-dimensional settings (e.g., p=1000 in Table 2 and Figure 3), its relatively higher false positive rates and less competitive predictive accuracy—compared to Bi-GMCPNet—can be attributed to well-known limitations of the LASSO penalty, which underlies the GLASSO framework. LASSO is known to introduce shrinkage bias, particularly for large true coefficients, and often includes noise variables as a compensatory effect [12,13]. This over-selection occurs in both low- and high-dimensional settings, but its impact is more severe when p≫n, where the risk of overfitting increases and predictive performance deteriorates [62,63]. As demonstrated in our simulations, this issue becomes especially relevant when tuning parameters are selected to optimize validation performance, as such criteria can favor overly complex models—a phenomenon also documented in theoretical studies of LASSO with cross-validation [64]. In contrast, the MCP penalty employed in our Bi-MCP and Bi-GMCPNet methods is specifically designed to reduce shrinkage bias and limit overselection [14,65]. Our simulation results consistently show that MCP-based approaches yield substantially lower false positive rates by more effectively distinguishing signal from noise in high-dimensional contexts. This leads to more parsimonious models and, consequently, improved predictive performance, as observed with Bi-GMCPNet.

Our investigations also extended to scenarios where the key assumption of identical predictor sets across outcomes was deliberately violated (Example 3). The results demonstrate the robustness of the proposed Bi-GMCPNet approach. Rather than limiting selection to the intersection of the true predictor sets ($S_{1}\cap S_{2}$), Bi-GMCPNet consistently identified the union ($S_{1}\cup S_{2}$), capturing variables associated with either failure type. This behavior is practically advantageous, as it produces a single parsimonious model that comprehensively reflects the relevant prognostic factors. Importantly, even under this assumption violation, the joint modeling framework allowed Bi-GMCPNet to borrow strength across outcomes, leading to improved variable detection—specifically, lower false negative rates for at least one outcome—compared to fitting separate models. This flexibility underscores the practical value of the method in multivariate survival analysis settings where predictor sets are partially overlapping rather than fully shared.

Historically, the disease state model has been widely applied to various types of cancer to provide a structured framework for understanding patient prognoses. This model outlines the patient journey from diagnosis through recurrence, progression to metastasis, and ultimately, death. Within this framework, the selection of appropriate outcomes is crucial, as different biological behaviors influence prognoses and clinical outcomes. Typically, clinical trials are designed with a primary endpoint tailored to each disease state. Commonly used endpoints in oncology include overall survival (OS) and composite endpoints such as progression-free survival (PFS) and disease-free survival (DFS), both of which incorporate death as a component.

In recent years, many randomized phase III trials have adopted co-primary endpoints. However, stratification in randomization is often based on one outcome (e.g., OS), which can lead to imbalances in baseline patient characteristics if stratification is based on prognostic factors for only one outcome, without accounting for others. Our focus is on understanding how multiple predictors simultaneously and consistently influence the timing of several clinically meaningful events. For example, understanding how predictors impact both OS and PFS is crucial for prognosis, treatment decision making, and drug development, especially in heterogeneous patient populations.

A key challenge lies in identifying prognostic factors by integrating information from multiple endpoints both within and across disease states. This requires modeling the potentially varying effects of prognostic variables across different outcomes. Our study addresses this challenge by proposing a novel method for analyzing multivariate failure time data. By identifying common prognostic factors across multiple outcomes, our approach can inform clinical trial design—particularly in selecting prognostic variables for stratification and identifying high-risk patients who may benefit from personalized treatment strategies. Although our motivation stems from composite outcomes that include death, our method is equally applicable to non-fatal endpoints and other time-to-event outcomes. Importantly, our approach differs from competing risk frameworks such as the cause-specific hazard model [66,67] or the Fine–Gray subdistribution hazard model [68]. Rather than treating death as a competing risk, our model treats it as an outcome of interest, thereby maximizing the relationship between multiple time-to-event outcomes.

One notable design choice of the proposed method is its exclusive use of group-level sparsity, which assumes that multiple time-to-event outcomes share a common set of relevant predictors. This assumption is motivated by the biological and clinical plausibility that distinct endpoints in complex diseases often arise from shared underlying mechanisms. For example, in prostate cancer, clinical variables such as ECOG performance status, disease stage, and time of diagnosis are known to influence both OS and PFS [69]. By encouraging joint variable selection, our approach enhances interpretability, improves statistical power, and simplifies computation. While this shared structure necessarily excludes within-group sparsity—i.e., selecting a variable for only a subset of the outcomes—it was a deliberate modeling strategy aimed at identifying predictors with consistent effects across endpoints. Our method enforces this structure by grouping model parameters associated with each variable across outcomes and applying a sparsity-inducing penalty that promotes between-group sparsity, thereby selecting or excluding entire variable groups simultaneously. Despite this limitation, our simulation studies under model misspecification (Example 3, Section 4.2) demonstrated that the method remains robust in identifying the union of relevant predictors—a property that is often sufficient and practically useful in multivariate survival analysis. For applications requiring outcome-specific variable selection, incorporating bi-level sparsity—achieving both group- and outcome-level selection—would provide additional flexibility. Developing such an extension within a nonlinear neural network framework poses notable methodological and computational challenges, making it a promising direction for future research. Potential approaches may involve adapting bi-level regularization techniques, such as the sparse group lasso [70] or composite penalties [41], to nonlinear modeling frameworks implemented via neural networks. Furthermore, establishing theoretical guarantees for our method, including selection consistency and convergence rates, remains an important future goal to strengthen its foundation.

Further limitations of the proposed methods also warrant consideration. While our approach achieves variable selection by inducing sparsity at the input layer, the nonlinear transformations within hidden layers can obscure interpretability compared to traditional linear models. Future work may explore strategies to enhance the transparency of these learned functions. Additionally, model performance is sensitive to architectural choices and hyperparameter tuning, particularly the group penalty (λ) and ridge penalty (α). Although we adopted a validation-based tuning strategy (Section 3), selecting optimal values and defining appropriate search ranges—such as λ spanning from a dense to an empty model—can be a non-trivial aspect of the modeling pipeline and may require iterative refinement. A potential direction for future work is to incorporate advanced hyperparameter selection techniques, such as the method proposed for high-dimensional Cox models in Sancar et al. (2023) [71], to further enhance our approach. Finally, in ultra-high-dimensional settings (e.g., hundreds of thousands of variables), as noted in Luo and Halabi (2023) [35], the proposed method may encounter scalability issues due to a complex optimization landscape. In such cases, preliminary feature screening may offer a practical way to reduce dimensionality prior to model fitting [65,72].

## 7. Conclusions

In this paper, we introduced and validated a novel regularized framework for simultaneous variable selection and model estimation in the context of multivariate failure time data, a common challenge in modern biomedical research. Our approach extends beyond traditional linear hazard models by first employing a penalized pseudo-partial likelihood method with group LASSO-type penalties (Bi-LASSO and Bi-MCP) for linear settings. We then significantly advanced this concept to capture complex, nonlinear predictor relationships using sparse-input feed-forward neural networks that share a common input layer and incorporate group-level penalization on input neuron weights (Bi-GLASSONet and Bi-GMCPNet).

Key findings from our extensive simulation studies demonstrated the superior performance of our proposed methods, particularly Bi-MCP for linear models and Bi-GMCPNet for nonlinear scenarios. These methods consistently achieved greater accuracy in variable selection, exhibiting lower false positive and false negative rates and enhanced predictive power compared to approaches applied to individual outcomes or standard LASSO-based group selection techniques. Notably, Bi-GMCPNet proved highly effective in high-dimensional settings and when handling intricate nonlinear associations, and it also showed remarkable robustness in identifying relevant predictors (capturing the union of influential variables) even when the assumption of perfectly identical predictor sets across outcomes was not met. The application of our framework to a real-world dataset from the CALGB-90401 advanced prostate cancer trial further underscored its practical utility. Bi-GMCPNet successfully identified established clinical prognostic factors and potentially important SNPs associated with both overall survival and progression-free survival, outperforming counterpart methods.

In summary, our novel method addresses key challenges in modeling individual survival outcomes. Its innovation lies in its ability to jointly model multiple endpoints, accounting for their interdependence while treating death as an outcome rather than a competing risk. The successful application of our method to both simulated and real-world data demonstrates its potential to advance prognostic research and improve clinical outcomes—not only in oncology but also in other complex disease areas. This comprehensive approach provides a powerful tool for identifying common prognostic factors, ultimately enhancing our understanding and management of disease through personalized treatment strategies.

## Figures and Tables

**Figure 1 bioengineering-12-00596-f001:**
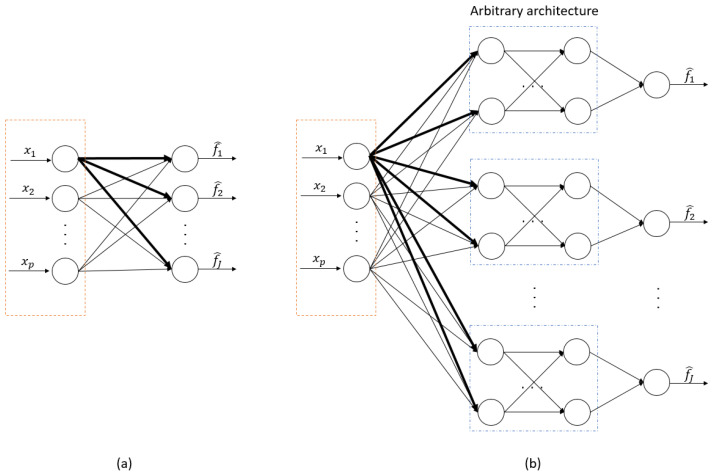
**The network architecture for (a) the marginal linear hazards model and (b) the marginal nonlinear hazards model**. In our proposed method, group regularization is applied to the $l_{2}$ norm of the weight vector associated with each input node. In each panel, the bolded connections show an example of a group for the first input node.

**Figure 2 bioengineering-12-00596-f002:**
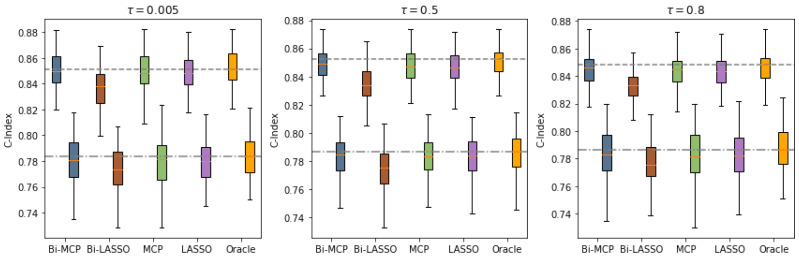
**C-index comparison for the linear bivariate failure time model (Example 1) under 30% censoring,** n=300**, and varying dependence levels (Kendall’s** τ=0.005**, 0.5, 0.8)**. Each panel corresponds to one τ value. For each method (Bi-MCP, Bi-LASSO, MCP, LASSO), two boxplots summarize the predictive performance across 200 replicates, one per outcome. The C-index quantifies how well a model ranks event times, with higher values indicating better discrimination. Dashed lines indicate the median C-index from the oracle method, representing ideal performance if true features were known.

**Figure 3 bioengineering-12-00596-f003:**
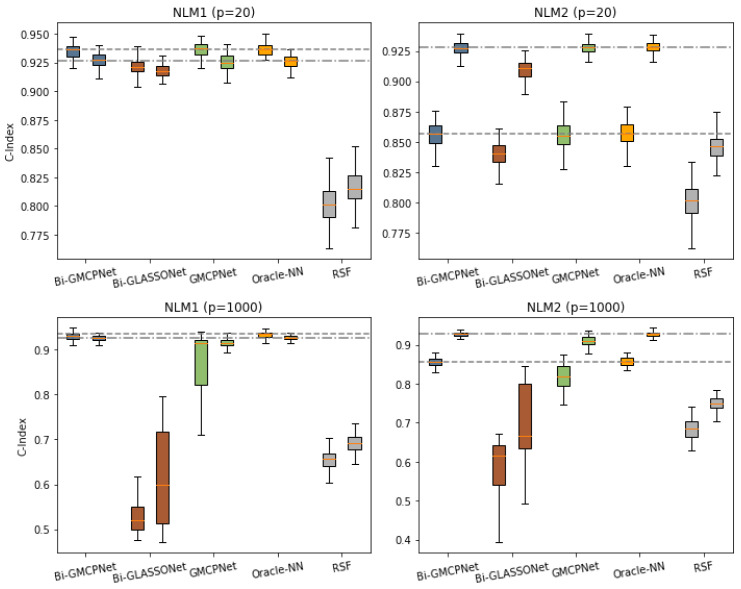
**C-index comparison for the nonlinear bivariate failure time model (Example 2) with Kendall’s** τ=0.5**.** The figure includes four panels, corresponding to two nonlinear models (NLM1 and NLM2), each evaluated under low-dimensional (*p* = 20) and high-dimensional (*p* = 1000) settings. Within each panel, boxplots show the C-index distributions across 200 replicates for each method, with two boxplots per method representing the two event outcomes. Dashed horizontal lines indicate the median C-index of the Oracle-NN method, which serves as a benchmark reflecting ideal predictive performance when the true underlying structure is known.

**Figure 4 bioengineering-12-00596-f004:**
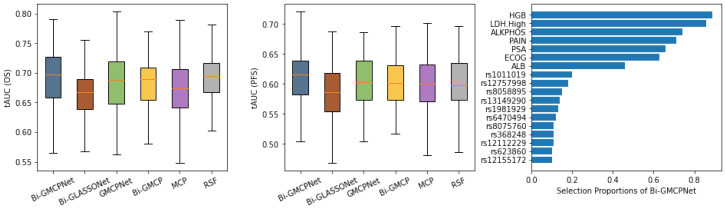
**Model performance and variable selection consistency on the CALGB-90401 prostate cancer dataset.** Results are based on 100 random train/test splits. **Left and middle panels:** boxplots showing the distribution of time-dependent AUC (tAUC) values for predicting overall survival (OS) and progression-free survival (PFS), respectively. Higher tAUC values indicate better predictive discrimination over time. Each method’s performance is summarized across the 100 splits. **Right panel:** variable selection stability of the Bi-GMCPNet model. Bars indicate the proportion of splits in which each clinical or SNP variable was selected (≥10%), reflecting robustness of feature selection across runs.

**Table 1 bioengineering-12-00596-t001:** **Feature selection results of Bi-MCP, Bi-LASSO, MCP, and LASSO for the linear hazards model outlined in Example 1.** The false positive rate (FPR %), false negative rate (FNR %), and model size (MS) with standard deviation (SD) in parentheses are displayed.

C	τ	Method	n=300				n=500			
			FPR	FNR	IR	MS	FPR	FNR	IR	MS
0.1	0.005	Bi-MCP	11.8	0.6	100	6.7 (3.3)	10.7	0.2	100	6.6 (3.3)
		Bi-LASSO	98.7	0.0	100	19.8 (0.4)	98.6	0.0	100	19.8 (0.5)
		MCP	20.1, 16.5	4.6, 28.8	0.0	7.8 (4.1), 6.0 (4.1)	16.7, 19.6	4.4, 24.2	0.0	7.3 (3.6), 6.7 (4.0)
		LASSO	45.7, 39.5	2.0, 18.2	0.0	11.8 (5.2), 10.0 (5.4)	43.3, 39.2	0.6, 17.8	0.0	11.5 (4.6), 10.0 (5.5)
	0.5	Bi-MCP	10.1	2.0	100	6.4 (3.4)	11.1	0.4	100	6.6 (2.9)
		Bi-LASSO	98.9	0.0	100	19.8 (0.4)	98.1	0.0	100	19.7 (0.6)
		MCP	20.3, 15.9	5.6, 28.2	0.0	7.8 (4.2), 6.0 (4.1)	19.2, 14.1	2.4, 28.4	1.0	7.8 (3.7), 5.7 (3.5)
		LASSO	43.5, 34.3	1.2, 20.0	1.0	11.5 (5.0), 9.2 (5.3)	48.0, 39.5	0.6, 19.4	0.0	12.2 (5.0), 10.0 (5.2)
	0.8	Bi-MCP	16.5	1.2	100	7.4 (4.2)	14.5	0.6	100	7.1 (4.1)
		Bi-LASSO	98.3	0.0	100	19.8 (0.5)	98.0	0.0	100	19.7 (0.6)
		MCP	24.1, 24.0	4.8, 26.2	1.0	8.4 (4.5), 7.3 (4.8)	15.7, 15.5	3.2, 23.8	1.0	7.2 (3.6), 6.1 (3.6)
		LASSO	46.0, 46.8	1.4, 16.2	0.0	11.8 (5.1), 11.2 (5.9)	43.7, 37.5	0.4, 16.4	0.0	11.5 (5.0), 9.8 (5.4)
0.3	0.005	Bi-MCP	9.3	2.0	100	6.3 (3.1)	13.7	0.6	100	7.0 (3.8)
		Bi-LASSO	99.3	0.0	100	19.9 (0.3)	98.8	0.0	100	19.8 (0.4)
		MCP	13.1, 18.2	8.2, 29.8	0.0	6.6 (3.6), 6.2 (4.4)	16.9, 16.7	3.8, 28.6	0.0	7.3 (3.7), 6.1 (3.9)
		LASSO	38.3, 41.9	2.4, 20.0	0.0	10.6 (4.7), 10.3 (5.4)	43.5, 38.4	0.6, 18.6	0.0	11.5 (4.8), 9.8 (5.5)
	0.5	Bi-MCP	13.3	1.0	100	6.9 (3.9)	10.8	1.2	100	6.6 (3.0)
		Bi-LASSO	99.2	0.0	100	19.9 (0.3)	98.3	0.0	100	19.8 (0.5)
		MCP	17.3, 16.3	7.4, 32.8	0.0	7.2 (3.9), 5.8 (4.1)	15.5, 16.1	3.8, 27.6	1.0	7.1 (3.4), 6.0 (3.7)
		LASSO	42.9, 35.5	3.6, 22.2	0.0	11.2 (5.0), 9.2 (5.3)	37.5, 35.3	0.8, 18.2	1.0	10.6 (4.5), 9.4 (4.9)
	0.8	Bi-MCP	12.7	1.2	100	6.8 (3.1)	13.4	0.6	100	7.0 (3.8)
		Bi-LASSO	99.3	0.0	100	19.9 (0.3)	98.6	0.0	100	19.8 (0.4)
		MCP	16.5, 15.7	6.8, 30.4	0.0	7.1 (3.6), 5.8 (3.7)	14.7, 16.7	4.0, 27.0	0.0	7.0 (3.4), 6.2 (4.2)
		LASSO	45.3, 38.9	1.8, 18.8	0.0	11.7 (5.2), 9.9 (5.4)	37.2, 38.7	1.0, 17.4	1.0	10.5 (4.6), 9.9 (5.3)

**Table 2 bioengineering-12-00596-t002:** **Feature selection results of Bi-GMCPNet, Bi-GLASSONet, and GMCPNet for the nonlinear hazards model outlined in Example 2.** The false positive rate (FPR %), false negative rate (FNR %), and model size (MS) with standard deviation (SD) in parentheses are displayed.

Model	τ	Method	p=20				p=1000			
			FPR	FNR	IR	MS (SD)	FPR	FNR	IR	MS (SD)
NLM1	0.005	Bi-GMCPNet	0.5	0.0	100	5.1 (0.4)	0.0	0.0	100	5.0 (0.0)
		Bi-GLASSONet	100.0	0.0	100	20.0 (0.0)	11.5	48.0	100	116.6 (194.1)
		GMCPNet	0.5, 1.1	2.2, 1.0	68.0	5.0 (0.4), 5.1 (0.6)	0.0, 0.0	24.8, 15.8	1.0	3.9 (0.6), 4.3 (0.4)
	0.5	Bi-GMCPNet	0.3	0.0	100	5.0 (0.5)	0.0	0.0	100	5.0 (0.1)
		Bi-GLASSONet	100.0	0.0	100	20.0 (0.0)	15.0	47.0	100	152.1 (262.0)
		GMCPNet	1.5, 2.7	1.0, 1.0	68.0	5.2 (1.5), 5.4 (1.7)	0.0, 0.0	25.2, 14.4	1.0	4.0 (0.9), 4.3 (0.5)
	0.8	Bi-GMCPNet	0.5	0.0	100	5.1 (0.4)	0.0	0.0	100	5.0 (0.1)
		Bi-GLASSONet	100.0	0.0	100	20.0 (0.0)	16.2	46.8	100	163.6 (277.6)
		GMCPNet	0.1, 1.9	2.2, 0.2	73.0	4.9 (0.4), 5.3 (0.7)	0.0, 0.0	24.4, 15.6	1.0	3.8 (0.7), 4.3 (0.4)
NLM2	0.005	Bi-GMCPNet	0.1	0.0	100	5.0 (0.1)	0.0	0.0	100	5.0 (0.2)
		Bi-GLASSONet	99.9	0.0	100	20.0 (0.1)	12.5	34.6	100	127.9 (140.1)
		GMCPNet	1.7, 1.0	0.0, 0.4	75.0	5.3 (0.8), 5.1 (0.8)	0.0, 0.0	10.6, 16.0	13.0	4.6 (0.7), 4.3 (0.7)
	0.5	Bi-GMCPNet	0.1	0.0	100	5.0 (0.1)	0.0	0.0	100	5.0 (0.1)
		Bi-GLASSONet	100.0	0.0	100	20.0 (0.0)	15.0	36.0	100	152.5 (216.9)
		GMCPNet	1.4, 0.4	0.2, 0.0	85.0	5.2 (0.7), 5.1 (0.2)	0.0, 0.0	14.0, 14.6	13.0	4.5 (0.9), 4.3 (0.6)
	0.8	Bi-GMCPNet	0.1	0.0	100	5.0 (0.2)	0.0	0.0	100	5.0 (0.1)
		Bi-GLASSONet	100.0	0.0	100	20.0 (0.0)	11.3	34.4	100	115.6 (138.2)
		GMCPNet	1.2, 0.5	0.2, 0.2	82.0	5.2 (0.6), 5.1 (0.4)	0.0, 0.0	13.8, 15.4	12.0	4.7 (1.6), 4.3 (0.6)

**Table 3 bioengineering-12-00596-t003:** **Feature selection results of Bi-GMCPNet, Bi-GLASSONet, and GMCPNet for the nonlinear hazards model with different predictor sets outlined in Example 3.** For each of the two failure types (outcome 1 and outcome 2), the table displays the false positive rate (FPR %), false negative rate (FNR %), model size (MS), and C-index. Standard deviations (SD) are provided in parentheses for MS and C-index.

*p*	Method	Outcome 1				Outcome 2			
		FPR	FNR	MS (SD)	C-index (SD)	FPR	FNR	MS (SD)	C-index (SD)
20	Bi-GMCPNet	14.0	0.0	7.1 (0.5)	0.848 (0.011)	14.0	0.0	7.1 (0.5)	0.923 (0.006)
	Bi-GLASSONet	100.0	0.0	20.0 (0.0)	0.822 (0.016)	100.0	0.0	20.0 (0.0)	0.902 (0.009)
	GMCPNet	1.5	0.0	5.2 (0.7)	0.854 (0.011)	0.8	0.0	5.1 (0.5)	0.928 (0.006)
1000	Bi-GMCPNet	0.2	11.2	6.5 (0.5)	0.820 (0.028)	0.1	0.0	6.5 (0.5)	0.923 (0.007)
	Bi-GLASSONet	26.4	48.5	265.4 (345.3)	0.548 (0.061)	26.4	40.5	265.4 (345.3)	0.628 (0.11)
	GMCPNet	0.0	13.5	4.5 (0.8)	0.820 (0.033)	0.0	14.7	4.4 (0.8)	0.910 (0.015)

## Data Availability

The clinical data from the CALGB 90401 can be accessed through the NCTN NCORP Data Archive.

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
