# Peer review of "Variable Selection for Multivariate Failure Time Data via Regularized Sparse-Input Neural Network"

_bioengineering, 2025, doi:10.3390/bioengineering12060596_

Round 1

Reviewer 1 Report

Comments and Suggestions for Authors

The manuscript entitled "Variable Selection for Multivariate Failure Time Data via Regularized Sparse-Input Neural Network” investigates the model estimation in multivariate failure time data analysis through identifying common predictors for multiple time-to-event endpoints in clinical trials. It is a well-written manuscript with supporting rules and equations; however, some points should be considered to improve the final quality of this manuscript as follows:

  1. Introduction: There is a complex relationship between predictors and time-to-event outcomes. Add a paragraph discussing the different types of these predictors.
  2. Some terms need to be explained with more details, like the problem of identifying variables pertinent to two time-to-event endpoints overall survival (OS) and progression-free survival (PFS)
  3. Discussion: Each method for estimation of multivariate failure time data encountered a number of limitations. The authors discussed one limitation for their model; however, they need to explain more details about the limitations that could be encountered by their model.
  4. Authors need to explain the benefits of application of their model against the other models
  5. Conclusion: Separate the summary part to be a conclusion and add some more details to conclude the main findings in this study
  6. References: The references list contains some old references like No. 41, 33... Replace them with new, well-established references.

Reviewer 2 Report

Comments and Suggestions for Authors

The manuscript is well structured. However, some issues need to be addressed before it can be published. 

The abstract must be rewritten in an organized manner according to the journal guidelines. 

The external and internal validation methods must be described in detail. The training set and test set must also be provided (or at least described in detail) as supplemental files. 

Figures 2 and 3 and 4  must be further described in order to increase visibility for a non-specialist reader. 

The discussion section must be expanded: 

- It is recommended that the method exclusively implements group-level sparsity. Although this is described as deliberate, the Discussion indicates that bi-level sparsity (i.e., both inter-group and intra-group) may be better suitable in numerous practical situations, yet it is not executed.

-As a recommendation, while a practical application (CALGB-90401 prostate cancer study) is demonstrated, the biological or clinical interpretation of the chosen SNPs and factors in Figure 4 (right panel, page 15) is not discussed in detail. Incorporating a discussion on the significance of these elements might improve the manuscript's quality. 

Reviewer 3 Report

Comments and Suggestions for Authors

Reviewer’s comments

The manuscript "Variable Selection for Multivariate Failure Time Data via Regularized Sparse-Input Neural Network" is well-written. The work is relevant, and the methodology is well-developed. However, some areas require significant revision to improve clarity and strengthen the manuscript so it reaches publication standards.

Here are my comments to improve the quality of the manuscript

Minor Corrections

  1. The assumption of a common set of predictors across all failure times is restrictive. While the authors discuss robustness to violations of this assumption (Example 3), a more explicit discussion of the implications and potential extensions (e.g., bi-level sparsity) would strengthen the manuscript.
  2. The manuscript would improve by including a clearer comparison with other advanced methods for high-dimensional survival analysis, like the ones mentioned in the suggested references (e.g., adaptive elastic net, two-stage feature selection). Including these comparisons would better contextualize the proposed method's advantages. The authors should consider citing the following studies to better situate their work within the literature: (Add: "A Two-Stage Feature Selection Approach Based on Artificial Bee Colony and Adaptive LASSO in High-Dimensional Data") for alternative feature selection strategies. (Add: "Adaptive elastic net based on modified PSO for Variable selection in cox model with high-dimensional data" for comparisons with adaptive penalization methods). (Add: "Enhanced MRI-based brain tumour classification with a novel Pix2pix generative adversarial network augmentation framework") for insights into advanced neural network applications in biomedical data.
  3. The availability of Python code is a positive aspect. However, the manuscript could include more details about the implementation, such as hyperparameter tuning (e.g., grid search ranges for "lambda" and "alpha") and computational complexity to facilitate reproducibility.
  4. Ensure all figures are included and clearly labelled in the provided manuscript.
  5. Could you please explain why Bi-GLASSONet does not perform well in high dimensions?

Round 2

Reviewer 2 Report

Comments and Suggestions for Authors

The authors have addressed all the comments.